# Effects of Deposition Power and Annealing Temperature on Indium Zinc Oxide (IZO) Film’s Properties and Their Applications to the Source–Drain Electrodes of Amorphous Indium Gallium Zinc Oxide (a-IGZO) Thin-Film Transistors (TFTs)

**DOI:** 10.3390/nano15110780

**Published:** 2025-05-22

**Authors:** Yih-Shing Lee, Chih-Hsiang Chang, Bing-Shin Le, Vo-Truong Thao Nguyen, Tsung-Cheng Tien, Horng-Chih Lin

**Affiliations:** 1Department of Semiconductor and Electro-Optical Technology, Minghsin University of Science and Technology, Hsinchu 30401, Taiwan; nguyenlemaitrinh1234@gmail.com (B.-S.L.); truongthaofpt2016@gmail.com (V.-T.T.N.); 2Department of Electronic Engineering, Minghsin University of Science and Technology, Hsinchu 30401, Taiwan; zvd66994652@yahoo.com.tw; 3Department of Materials Science and Engineering, National Yang Ming Chiao Tung University, Hsinchu 30010, Taiwan; ttt123175721@yahoo.com.tw; 4Institute of Electronics, National Yang Ming Chiao Tung University, 1001 Ta-Hsueh Road, Hsinchu 300, Taiwan; hclin@nycu.edu.tw

**Keywords:** RF sputtering power, annealing temperature, IZO source-drain (S–D) electrodes, total resistance method, a-IGZO TFTs

## Abstract

The optical, electrical, and material properties of In–Zn–O (IZO) films were optimized by adjusting the deposition power and annealing temperature. Films deposited at 125 W and annealed at 300 °C exhibited the best performance, with the lowest resistivity (1.43 × 10^−3^ Ω·cm), highest mobility (11.12 cm^2^/V·s), and highest carrier concentration (4.61 × 10^20^ cm^−3^). The average transmittance and optical energy gap were 82.57% and 3.372 eV, respectively. The electrical characteristics of amorphous In-Ga-Zn-O (a-IGZO) thin-film transistors (TFTs) using IZO source-drain (S–D) electrodes with various sputtering powers and annealing temperatures were investigated. The optimal sputtering power of 125 W and annealing temperature of 300 °C for the IZO S–D electrodes resulted in the highest field-effect mobility (~12.31 cm^2^/V·s) and on current (~2.09 × 10^−6^ A). This improvement is attributed to enhanced carrier concentration and mobility, which result from the high In/Zn ratio, the larger grain size, and low RMS roughness in the IZO films. The parasitic contact resistance (*R_SD_*) and channel resistance (*R_CH_*) were analyzed using the total resistance method. *R_SD_* decreased with increasing IZO S–D sputtering power, while *R_CH_* reached a minimum at 125 W. Both resistances decreased significantly as the annealing temperature increased from 200 °C to 300 °C.

## 1. Introduction

Indium gallium zinc oxide (IGZO) thin-film transistors (TFTs) are highly attractive for display applications due to their high transparency and excellent electrical properties, including a high mobility and a high on/off current ratio. These characteristics make them particularly appealing for use in brain-like synaptic transistors, back planes in CMOS image sensors, integrated circuits (ICs), flexible and wearable electronics, power switching circuits, BEOL transistor elements in 3D logic, and cell transistors in DRAM [1,2,3,4,5]. In general, the electrical performance of oxide semiconductor TFTs is related to the contact resistance between the channel layer and the source-drain (S–D) electrode materials, as well as the channel resistance [6,7,8,9,10]. Shimura et al. [6] used various contact materials, including silver (Ag), indium (In), titanium (Ti), indium tin oxide (ITO), and amorphous indium zinc oxide (a-IZO), to fabricate the contact layers of the devices and study their impact on contact resistance (ρ_C_) under different processing conditions. The TFTs with ITO and Ti as the source and drain contact layers showed better device performance than those with gold (Au) Schottky contacts. Barquinha et al. [7] explored the possibility of replacing ITO with different materials in inverted staggered TFTs, which use IGZO as the semiconductor material. The electrode materials analyzed include IZO, Ti, aluminum (Al), molybdenum (Mo), and titanium/gold (Ti/Au). Each material was applied in two different devices: one where annealing was performed after the IGZO channel deposition and before the S–D deposition, and the other where annealing was performed at the end of the device fabrication. The results showed that the electrical characteristics improved when annealing was performed at the end of the device fabrication. Due to the potential formation of energy barriers at the interface between the IGZO channel layer and ITO thin films, when ITO is used as the S–D electrode for IGZO TFTs, an interface resistance may form between the two layers, affecting the device performance. Choi et al. [10] proposed an innovative design by co-sputtering IGZO and ITO mixed films at room temperature (with a thickness of 125 nm), followed by the deposition of a 375 nm thick ITO layer as the S–D electrode to fabricate TFTs. The proposed TFT demonstrates enhanced electrical performance, attributed to its lower contact and channel resistance, as well as the formation of a surface reaction between the S–D and channel layers, which generated additional oxygen vacancies in the IGZO channel region.

IZO thin films with excellent optical and electrical properties have been widely utilized in oxide thin-film transistors, including as materials for the source and drain regions, channel layers, and top gate electrodes of the dual-gate devices, by adjusting the reactive oxygen content in the sputter chamber during deposition, as well as by varying the annealing temperatures and atmospheres [11,12,13,14,15,16,17,18]. Research on IZO thin films primarily utilizes radio-frequency (RF) and DC magnetron sputtering technology and covers various processing conditions, such as different power, process pressure, oxygen flow rates, and substrate temperatures, as well as various annealing temperatures and variations in the annealing atmosphere [19,20,21,22,23]. The effects of various processing parameters on the structural, optical, and electrical characteristics of the IZO thin films have been studied. However, there has been limited research on the effects of IZO S–D electrode processes with varying deposition powers and annealing temperatures on the electrical characteristics of a-IGZO TFTs.

This study investigates the optical, electrical, and material properties of IZO films deposited at different RF magnetron sputtering powers under Ar and O_2_ atmospheres at room temperature. Following deposition, the films were annealed at various temperatures in an N_2_ atmosphere. Moreover, an inverted–staggered transistor structure was used on silicon wafers to fabricate IGZO TFTs in this work. The effects of IZO S–D electrode processes with varying sputtering powers and annealing temperatures on the transfer characteristics of a-IGZO TFTs were studied. Output characteristics of devices with different channel lengths (*L*) were analyzed using the total resistance method [24,25]; the channel resistance (*R_CH_*) and S–D contact resistance (*R_SD_*) at different sputtering powers and annealing temperatures of IZO were extracted and analyzed. The IGZO TFT device with an IZO electrode deposited at 125 W and annealed at 300 °C exhibited the highest field-effect mobility (*μ_FE_*) and on current (*I_on_*). This enhancement is attributed to the IZO film having the highest In/Zn atomic ratio, the largest average grain size, and the smoothest surface, as measured by energy-dispersive spectroscopy (EDS), X-ray diffraction (XRD), and atomic force microscopy (AFM), respectively. These properties lead to increased carrier mobility and concentration in the IZO films, resulting in reductions in the total resistance (*R_total_*), contact resistance (*R_SD_*), and channel resistance (*R_CH_*) in a-IGZO TFTs with IZO S–D electrodes. This study aims to establish a causal relationship between the properties of IZO films and the characteristics of the source–drain electrodes in a-IGZO TFTs.

## 2. Materials and Methods

### 2.1. IZO Film Deposition and Characterization Measurements

IZO thin films were deposited at room temperature on B270 glass substrates and Si wafers using RF magnetron sputtering with a 7.62 cm (3-inch) diameter IZO target (In_2_O_3_: ZnO = 90 wt%: 10 wt%, 99.99% purity). The sputtering power was varied to 75 W, 100 W, 125 W, and 150 W. Prior to deposition, the chamber was evacuated to a base pressure of 5.3 × 10^−4^ Pa. Argon (50 sccm) and oxygen (1 sccm) gases were then introduced to maintain a working pressure of 0.67 Pa during the process. Following deposition, the films were annealed at 200 °C, 300 °C, and 350 °C for 1 h in a nitrogen (N_2_) ambient (40 sccm) condition at the same working pressure. Film resistivity, carrier mobility, and carrier concentration were determined using a Hall measurement system (HALL 8800, Swin, Hsinchu, Taiwan). The surface morphology and chemical composition of the IZO thin films were characterized using an ultra-high-resolution field-emission scanning electron microscope (FE-SEM, JEOL JSM-7800F Prime, Akishima, Tokyo, Japan) equipped with an energy-dispersive spectrometer (EDS, Oxford MAX150, High Wycombe, UK). Crystallinity was examined via grazing incidence X-ray diffraction (GIXRD) using a Bruker D8 Discover system (Billerica, MA, USA) with a Ni-filtered Cu Kα radiation source (λ = 1.5418 Å) at a fixed incident angle of 0.7°. Scans were conducted over a 2θ range of 20–70° at a step size of 0.02°/s. Surface roughness was evaluated using an atomic force microscope (AFM, Nanoview 1000, UTEC Material, Taipei, Taiwan) over a 5 µm × 5 µm scan area.

### 2.2. Thin-Film Transistor Fabrication and Characterization Measurements

In this study, an inverted-staggered thin-film transistor (TFT) structure was employed for the fabrication of a-IGZO TFT devices. A schematic cross-sectional view of the device is shown in Figure 1. Detailed fabrication procedures are described elsewhere [25]. The IGZO active layer, with a thickness of 50 nm, was deposited at room temperature via RF magnetron sputtering using a target with an atomic composition of In_2_O_3_:Ga_2_O_3_:ZnO = 1:1:2. The sputtering process was carried out at a constant power of 75 W, with Ar/O_2_ gas flow rates maintained at 50/1 sccm. The system’s background pressure and processing pressure were 5.3 × 10^−4^ Pa and 0.67 Pa, respectively. The sputtering power of the IZO films and the annealing temperatures in an N_2_ atmosphere at the end of the device fabrication were varied to investigate the effects of IZO films as source and drain electrodes on the electrical properties of IGZO TFTs. The devices were annealed at 200 °C, 300 °C, and 400 °C under the same working pressure, N_2_ flow rate, and duration as used for the IZO film conditions. Electrical measurements were carried out using an Agilent 4156A precision semiconductor parameter analyzer (Santa Clara, CA, USA) at a controlled temperature of 25 °C. For the transfer characteristics (*I_DS_*-*V_GS_*), *V_GS_* was varied from −10 V to + 20 V in 0.3 V steps for *V_DS_* = 0.1 V and 10 V, respectively. The calculation methods for *µ_FE_* and the subthreshold swing (*S.S.*) are detailed in a previous study [26]. For the output characteristics (*I_DS_*–*V_DS_*), *V_DS_* was varied from 0 V to + 20 V in 0.2 V steps for *V_GS_*–*V_th_* = 5–11 V in the tested devices. The channel width (W) was fixed at 400 µm, and the designed channel length (*L*), defined as the distance between the source and drain metal pads, was varied from 10 µm to 100 µm. For a low *V_DS_* (0.1 V), *R_total_* as a function of the designed L was evaluated using the total resistance method, applied within the linear region of the device’s output characteristics [24,25].(1)$$R_{total}=\frac{V_{DS}}{I_{DS}}=R_{ch}+R_{SD}=r_{ch}\left(L-∆L\right)+R_{SD}$$(2)$$r_{ch}=\frac{1}{\mu_{E}C_{OX}W(V_{GS}-V_{th})}$$
where *r_ch_* denotes the channel resistance per channel length; *R_CH_* and *R_SD_* represent the channel resistance and the S–D contact resistance, respectively; *C_ox_* is the oxide capacitance per unit area; *W* is the channel width and *L* is the designed channel length, with ∆*L = L* − *L_eff_*, where *L_eff_* is the effective channel length; the threshold voltage (*V_th_*) is defined as the *V_GS_* at which the drain current (*I_DS_*) reaches a fixed value of 10^−9^ × (*W*/*L*) A; and *µ_E_* represents the effective mobility. Further details about the total resistance method can be found in a previous study [25].

## 3. Results and Discussion

### 3.1. Properties of IZO Films at Various Sputtering Powers and Annealing Temperatures

Figure 2a,b show the transmittance spectra and optical bandgap of IZO films deposited with different sputtering powers and annealed at 300 °C, respectively. In this study, we carefully adjusted the deposition time to control the thickness of the IZO films in the range of 199–205 nm at various sputtering powers, based on the deposition rate of our RF sputtering system. The effect of thickness variation on the average transmittance (T_avg_) was not significant due to the thickness variation being less than 5% in this study. As the sputtering power increased, the transmittance spectra of IZO thin films shifted toward longer wavelengths. Figure 2a shows that T_avg_ in the wavelength range of 400 to 800 nm for IZO films deposited at 125 W and annealed at 300 °C increased to 82.57%, whereas the T_avg_ of IZO films deposited at 150 W slightly decreased to 81.59%. In a direct bandgap semiconductor, the absorption coefficient (α) and the optical bandgap (*E_g_*) can be estimated using the standard Tauc plot method from the optical transmission spectra, as described in references [27,28]:(3)$${\left(\alpha h\nu \right)}^{2}=B\left(h\nu -Eg\right)$$(4)$$\alpha =2.303\frac{log\left(\frac{1}{T}\right)}{d};hv=hc/\lambda$$
where *B* is a material-dependent constant, *λ* is the wavelength of the incident photon, *hν* is the photon energy, *T* is the transmittance, and *d* is the thickness of the IZO films. According to the combined Equations (3) and (4), a plot is created with the y-axis representing (*αhν*)^2^ and the x-axis representing *hν*. After generating the curve based on the above data, a linear regression is performed using the last five data points. The intercept of the fitted line with the x-axis corresponds to the optical bandgap (*E_g_*) of the IZO films at various sputtering powers after annealing in N_2_ at 300 °C, as shown in Figure 2b. The *E_g_* values of the IZO films deposited at powers of 75, 100, 125, and 150 W and annealed at 300 °C were 3.19, 3.29, 3.37, and 3.24 eV, respectively. When the sputtering power increased from 75 W to 125 W, the optical bandgap increased from 3.19 eV to 3.37 eV; however, the optical bandgap of the IZO film deposited at 150 W decreased to 3.24 eV.

Figure 3a–c depict the Hall resistivity, carrier mobility, and carrier concentration of IZO films at different sputtering powers and various annealing temperatures. For the pristine IZO films, the resistivity was too high for Hall measurements and was therefore measured using a four-point probe method. As the sputtering power increased from 75 W to 125 W, the resistivity of the IZO films gradually decreased, while carrier mobility and carrier concentration increased. However, the resistivity of IZO films deposited at 150 W showed a slight increase, while both carrier mobility and carrier concentration significantly decreased. Figure 3d shows the Hall characteristics of IZO films deposited at 125 W under different annealing temperatures. As the annealing temperature increased from room temperature to 300 °C, the resistivity of the IZO films significantly decreased, while carrier mobility reached its maximum at 125 W, and carrier concentration peaked at 350 °C. However, when the annealing temperature reached 350 °C, the resistivity showed a slight increase, and carrier mobility slightly decreased. Therefore, the optimal electrical properties for IZO films were achieved at a sputtering power of 125 W and an annealing temperature of 300 °C, resulting in a resistivity of 1.73 × 10^−3^ Ω·cm, a carrier mobility of 11.12 cm^2^/V·s, and a carrier concentration of 4.61 × 10^20^ cm^−3^. It is worth mentioning that the resistivity of a-IGZO films at room temperature and annealing temperatures of 200 °C, 300 °C, and 400 °C were 1.25 × 10^4^, 7.81 × 10^3^, 0.218, and 2.10 × 10^−2^ Ω·cm, respectively, as measured by the four-point probe. As the annealing temperature increases, the resistivity of the a-IGZO film gradually decreases.

Figure 4a shows the In, Zn, and resulting In/Zn atomic ratios of IZO films deposited at various powers and annealed at 300 °C. When the sputtering power increased from 75 W to 125 W, the indium (In) atomic percentage increased, whereas the zinc (Zn) atomic percentage decreased, resulting in a clear increase in the In/Zn ratio from 3.72 to 4.22. However, when the sputtering power reached 150 W, the In atomic percentage decreased, while the Zn atomic percentage increased, resulting in an abrupt decrease in the In/Zn ratio of 3.47 in the film. Figure 4b shows the In, Zn, and resulting In/Zn ratios in the films deposited at 125 W as a function of annealing temperature. The results were obtained using the EDS technique. While the comparison was only relative across different conditions, it clearly demonstrated that the In, Zn, and resulting In/Zn atomic ratios in the deposited films could be effectively controlled by adjusting the sputtering power of the IZO target and the annealing temperature. As the annealing temperature increased from 200 °C to 350 °C, the In atomic percentage continued to increase, while the Zn atomic percentage steadily decreased, leading to a continuous increase in the In/Zn ratio from 3.53 to 4.33, as shown in Figure 4b. Therefore, the In/Zn ratio in IZO films reached its maximum value of 4.33 at a sputtering power of 125 W and an annealing temperature of 350 °C. Figure 4a,b shows that the trend of the In/Zn ratio changed with increasing sputtering power and annealing temperature, respectively—a trend consistent with the carrier concentration variation shown in Figure 3c,d. These results indicate that the carrier concentration of the IZO film is related to the In/Zn ratio in the composition of the film. Sufficient doping of In^3^⁺ with Zn^2^⁺ in IZO films promotes the formation of oxygen vacancies. These vacancies are considered the primary source of the n-type free electrons in the film [21,29,30], leading to an increase in carrier concentration.

Figure 5a,b show the XRD crystalline phase analysis of IZO thin films at an annealing temperature of 300 °C with different sputtering powers and at a sputtering power of 125 W with different annealing temperatures, respectively. The crystalline phase analysis of the IZO thin film was conducted using GIXRD. The peak positions of In_2_O_3_ include a broad amorphous peak corresponding to the (222) plane at approximately 2θ ≈ 32.0–32.6° and a distinctive peak corresponding to the strongest crystalline plane, (440), at 2θ ≈ 52.0–52.5°, as indicated by JCPDS No. 76-0152. These peaks are observed in the diffraction patterns of IZO films, as shown in Figure 5a,b. The average grain size (D) was estimated from the full width at half maximum (FWHM) of the In_2_O_3_ (440) diffraction peak using the Scherrer equation [31].D = 0.9λ/βcosθ(5)
where λ is the X-ray wavelength (λ = 0.15418 nm), θ is the Bragg diffraction angle, and β is the full width at half maximum (FWHM) of the diffraction peak, expressed in radians. Using Equation (5), the average grain sizes (D values) of IZO films annealed at 300 °C were calculated to be 31, 34, 55, and 40 nm for sputtering powers of 75, 100, 125, and 150 W, respectively. As shown in Figure 6, the D values were 34, 55, and 44 nm at annealing temperatures of 200, 300, and 350 °C, respectively. The IZO film deposited at 125 W and annealed at 300 °C exhibited the largest grain size, which corresponded to the highest carrier mobility, as shown in Figure 3b. Furthermore, as shown in Figure 3d, carrier mobility increased with annealing temperature from 200 °C to 300 °C, attributed to grain growth and the corresponding reduction in grain boundary scattering. However, a further increase in annealing temperature to 350 °C resulted in decreased mobility, likely due to enhanced grain boundary and ionized impurity scattering. These results indicate that optimizing sputtering power and annealing temperature can effectively control the crystallinity of the In_2_O_3_ (440) phase and minimize grain boundary scattering in IZO films.

To explain the effects of different sputtering powers on the IZO film’s optical and electrical properties, the surface roughness of the film was further analyzed. A 500 nm SiO_2_ film was deposited on a silicon substrate, with an RMS roughness of 4.2 nm and an Ra of 3.21 nm, serving as the substrate layer for the AFM sample. Figure 6a–d show AFM images, RMS roughness, and Ra average roughness of IZO films deposited at different powers annealed at 300 °C. At 300 °C annealing, when the IZO sputtering power increased from 75 W to 125 W, the surface roughness RMS decreased from 5.2 nm to 4.66 nm and the Ra decreased from 4.02 nm to 3.66 nm, respectively. However, at a sputtering power of 150 W, the surface roughness in terms of RMS and Ra slightly increased to 4.8 nm and 3.91 nm, respectively. From the AFM surface roughness results of the IZO film, as the sputtering power increased from 75 W to 125 W, the surface roughness significantly decreased, leading to an increase in carrier mobility from 4.49 to 11.12 cm^2^/V·s. However, when the sputtering power increased to 150 W, the slight increase in surface roughness caused the carrier mobility to decrease to 10.2 cm^2^/V·s. The trends of RMS and Ra surface roughness of the IZO films as a function of sputtering power show an inverse relationship with the average grain size estimated using the Scherrer equation. Ku et al. [19] reported that the mobility values of the IZO films deposited with oxygen contents of 0.5% and 1.0% decreased upon crystallization, whereas those deposited with 0.25% oxygen content continued to increase. These differences arise from variations in their crystalline structures, specifically the presence of the In_2_O_3_ crystalline phase in the crystallized films. From the optical transmittance spectrum results in Figure 2a, at a sputtering power of 150W, the increase in surface roughness also caused a slight decrease in average transmittance. In summary, the composition and microstructure of the IZO film have a significant impact on the optical and electrical properties of the transparent conductive film.

The effects of sputtering power and annealing temperature of IZO films, used as source and drain electrodes, on the electrical characteristics of a-IGZO TFTs are discussed in the following sections.

### 3.2. Electrical Characteristics of A-IGZO TFTs Using IZO Films as S–D Electrodes with Various Deposition Powers

Figure 7 shows the *I_DS_*-*V_GS_* and *I_DS_*^1/2^-*V_GS_* transfer characteristic curves of a-IGZO TFTs at W/L = 400/100 μm and an annealing temperature of 300 °C with IZO as the S/D electrodes deposited at different powers characterized at *V_DS_* = 0.1 V. Measurements were conducted on more than three samples per device. The gate capacitance per unit area (*C_ox_*) of the 100 nm TEOS oxide was measured to be 4.98 × 10^−8^ F/cm^2^. Table 1 presents the average values and standard deviations of the extracted electrical parameters of a-IGZO TFTs with IZO electrodes deposited at varying powers. The subthreshold swing (*SS*) value approximately doubles, ranging from ~0.62 to ~1.22 V/decade as the sputtering power increases. This is due to the increased deep-level defects in the a-IGZO channel layer caused by Ar ion bombardment [23]. Additionally, *V_th_*, ranging from ~−0.24 to ~2.11 V, and the tangent voltage of the *I_DS_*^1/2^ vs. *V_GS_* curve shift toward more positive values, which is related to the increase in deep-level defects within the channel layer and interface defects at the dielectric layer. The field-effect mobility (*μ_FE_*) (~12.31 cm^2^/V·s) and the on current (~2.09 × 10^−6^ A) reach their maximum values at a sputtering power of 125 W. This is attributed to the highest In/Zn atomic ratio observed at 125 W, which leads to an increase in the carrier concentration of the IZO films, as shown in Figure 3c. In addition, the largest average grain size (D) and the lowest RMS roughness at 125 W result in the highest carrier mobility, as shown in Figure 3b. Consequently, these factors contribute to the lowest film resistivity of the IZO films at 125 W, as shown in Figure 3a. However, the IZO films deposited at 150 W exhibit lower carrier mobility and carrier concentration, leading to higher film resistivity, as shown in Figure 3a–c, and resulting in a slight decrease in *μ_FE_* (~9.89 cm^2^/V·s) and on-current (*I_on_*, ~1.82 × 10^−6^ A) of the a-IGZO TFTs.

Figure 8a–c show the *I_DS_*-*V_DS_* output characteristic curves of a-IGZO TFTs at W/L = 400/100 μm and an annealing temperature of 300 °C with IZO as the S/D electrodes deposited at different powers characterized at various gate overdrive *V_GS_* − *V_th_* = 5–11 V. When *V_GS_* − *V_th_* = 11 V and the IZO sputtering powers are 75 W, 125 W, and 150 W, the corresponding saturation drain currents are approximately 2.88 × 10^−5^, 4.09 × 10^−5^, and 3.90 × 10^−5^A, respectively. From the *I_DS_*-*V_DS_* curves, it can be observed that the saturation current no longer increases for a power of 125 W. This result may be attributed to the higher In/Zn ratio and carrier concentration (~5.77 × 10^20^ cm^−3^) in the IZO film at 125W, as shown in Figure 4a and Figure 3c, respectively. Figure 9a–c show the relationship between *R_total_* and different channel lengths (R-L plots) for a-IGZO TFT devices (W = 400 μm) with IZO deposited at different powers and annealed at 300 °C characterized at *V_DS_* = 0.1 V and *V_GS_* − *V_th_* = 5–11 V. Using Equation (1), the *R_total_*, *R_SD_*, and *R_CH_* values for each channel length can be extracted (where Δ*L* = 0 μm for 75 W and 150 W). Additionally, for the 125 W power condition, an intersection appears when applying the total resistance method, yielding *R_SD_* = 1 × 10^4^ Ω, Δ*L* = 3.5 μm, as shown in the inset of Figure 9b. The channel length shrinkage, Δ*L* = 3.5 μm, can be attributed to the critical-dimension loss of the S/D electrode spacing, which occurs during the lithographic and subsequent lift-off process steps. Figure 10a–c show the extracted *R_total_*, *R_CH_*, and *R_SD_* at various gate overdrives for a-IGZO TFTs with IZO electrodes (L = 100 μm) deposited at powers of 75 W, 125 W, and 150 W and annealed at 300 °C. Based on the R–L plots in Figure 9, with a W/L ratio of 400/100 μm and a gate overdrive of 9 V, increasing the sputtering power from 75 W to 150 W results in a slight decrease in *R_CH_* from 1.73 × 10^5^ Ω to 1.43 × 10^5^ Ω, while *R_SD_* decreases significantly from 1.32 × 10^4^ Ω to 3.93 × 10^3^ Ω. This improvement is attributed to the reduced resistivity of the IZO films at higher sputtering powers, as shown in Figure 3a. IZO films deposited at higher powers exhibit larger average grain sizes (Figure 5) and lower RMS surface roughness (Figure 6), which in turn reduce *R_SD_* at the interface between the IZO electrodes and the a-IGZO channel layer. According to Equation (1) at *V_GS_* − *V_th_* = 11 V, as the sputtering power increases from 75 W to 125 W, *R_CH_* decreases from 1.30 × 10^4^ Ω to 8.50 × 10^3^ Ω. However, at 150 W, *R_CH_* increases to 1.14 × 10^4^ Ω. Since *R_CH_* reaches its lowest value at a sputtering power of 125 W across all gate overdriving voltages, Equation (2) indicates that this results in the highest effective mobility (*μ_E_*). Based on these electrical characteristics, the optimal sputtering power for IZO source–drain electrodes in a-IGZO TFTs is determined to be 125 W.

### 3.3. Electrical Characteristics of A-IGZO TFTs Determined by Using IZO Films as S–D Electrodes with Various Annealing Temperatures

Figure 11 shows the *I_DS_*-*V_GS_* and *I_DS_*^1/2^-*V_GS_* transfer characteristics of a-IGZO TFTs at *V_D_*_S_ = 0.1 V and W/L = 400/100 μm with IZO as the S–D electrodes deposited at a power of 125 W and annealed at different temperatures in ambient N_2_. Table 2 shows the extracted parameters of a-IGZO TFT devices with IZO electrodes deposited at a power of 125 W and annealed at different temperatures. As the annealing temperature increases from the pristine state to 300 °C, the *S.S.* value decreases by approximately half (from 1.88 to 0.85 V/decade), while the *μ_FE_* value increases from 4.71 to 12.31 cm^2^/V·s and the on current rises from 4.69 × 10^−7^ to 2.09 × 10^−6^ A. This improvement is due to high-temperature annealing, which significantly improves the interface between the a-IGZO channel layer and the dielectric layer, reducing thin-film trap defects and increasing film density [32]. Additionally, the threshold voltage decreases from 6.6 to 0.33 V, and the *I_DS_*^1/2^ vs. *V_GS_* curve clearly shows a shift in the tangent voltage toward a more negative voltage. This shift is attributed to high-temperature annealing, which increases the carrier concentration in the a-IGZO channel layer and decreases the film resistivity. Furthermore, annealing at 400 °C leads to depletion-mode TFTs due to the excessively low channel resistivity (2.10 × 10^−2^ Ω·cm) of a-IGZO films, as measured by the four-point probe, and the higher *I_off_* current (8.65 × 10^−6^ A). As a result, the *V_th_*, *SS*, and *μ_FE_* parameters cannot be determined.

To establish the relationship between IZO S–D electrodes annealed at different temperatures and device performance, the output characteristics of devices with varying channel lengths (L) were analyzed using the total resistance method. From the *R–L* plots for a-IGZO TFT devices (W = 400 μm) with IZO as the S–D electrodes annealed at different temperatures, characterized at *V_DS_* = 0.1 V and *V_GS_* − *V_th_* = 5–11 V, the *R_total_*, *R_SD_*, and *R_CH_* values for each channel length can be extracted using Equation (1). Figure 12a,b show the *R_total_*, *R_SD_*, and *R_CH_* plots at different gate overdrive voltages for IGZO TFTs with IZO as the S–D electrodes annealed at 200 °C and 300 °C, respectively. The source–drain contact resistance (*R_SD_*) decreases significantly—by nearly an order of magnitude—as the annealing temperature increases from 200 °C to 300 °C. This result is attributed to the increased carrier concentration and mobility in the IZO film with rising annealing temperature, as shown in Figure 3d, which leads to a marked reduction in both film resistivity and *R_SD_*. Furthermore, as the annealing temperature increases from 200 °C to 300 °C, both *R_total_* and *R_CH_* decrease significantly by a factor of 2 to 3. This is due to the reduced resistivity of the IZO film, as illustrated in Figure 3d, and the substantial decrease in the resistivity of the a-IGZO thin films—from 7.81 × 10^3^ to 0.218 Ω·cm. Therefore, the optimal annealing temperature for IZO as the source–drain electrodes in a-IGZO TFTs is 300 °C.

## 4. Conclusions

In summary, the optimal electrical performance of IZO thin films was achieved at a sputtering power of 125 W and an annealing temperature of 300 °C, where the films exhibited the lowest resistivity, the highest carrier mobility, and enhanced carrier concentration. These improvements are attributed to increased grain size, reduced surface roughness, and a favorable In/Zn atomic ratio. Further increases in sputtering power (to 150 W) or annealing temperature (to 350 °C) led to slight degradation in mobility and resistivity, likely due to increased grain boundary and ionized impurity scattering.

The *μ_FE_* (~12.31 cm^2^/V·s) and the on current (~2.09 × 10^−6^ A) reach their maximum values at a sputtering power of 125 W. This is attributed to the highest In/Zn atomic ratio observed at 125 W, which leads to an increase in the carrier concentration of the IZO films. In addition, the largest average grain size and the lowest RMS roughness at 125 W result in the highest carrier mobility. Consequently, these factors contribute to the lowest film resistivity of the IZO films at 125 W. However, the IZO films deposited at 150 W exhibit lower carrier mobility and carrier concentration, leading to higher film resistivity, and resulting in a slight decrease in the *μ_FE_* (~9.89 cm^2^/V·s) and on current (*I_on_*, ~1.82 × 10^−6^ A) of the a-IGZO TFTs. According to the total resistance method, IZO films deposited at higher powers exhibit larger average grain sizes and lower RMS surface roughness, which in turn reduce *R_SD_* at the interface between the IZO electrodes and the a-IGZO channel layer. Additionally, *R_CH_* reaches its minimum value at a sputtering power of 125 W across all gate overdrive voltages, indicating that this results in the highest effective mobility (*μ_E_*) in the channel. Based on these electrical characteristics, the optimal sputtering power for IZO source–drain electrodes in a-IGZO TFTs is determined to be 125 W.

As the annealing temperature increases from the pristine state to 300 °C, the *SS* value decreases by approximately half, while the *μ_FE_* and *I_on_* current gradually increase. This improvement is attributed to high-temperature annealing. The *V_th_* clearly shifts toward negative voltage. The *R_SD_* decreases significantly—by nearly an order of magnitude—as the annealing temperature increases from 200 °C to 300 °C. This result is attributed to the increased carrier concentration and mobility in the IZO film with rising annealing temperature, which leads to a marked reduction in both film resistivity and *R_SD_*. Furthermore, as the annealing temperature increases from 200 °C to 300 °C, both *R_total_* and *R_CH_* decrease significantly by a factor of 2 to 3. This is due to the reduced resistivity of the IZO film and the substantial decrease in the resistivity of the a-IGZO thin films. Therefore, 300 °C is identified as the optimal annealing temperature for IZO as the S–D electrodes in a-IGZO TFTs. Based on the analysis of electrical properties of a-IGZO TFTs with IZO source–drain electrodes, the optimal sputtering power and annealing temperature were determined to be 125 W and 300 °C, respectively.

## Figures and Tables

**Figure 1 nanomaterials-15-00780-f001:**
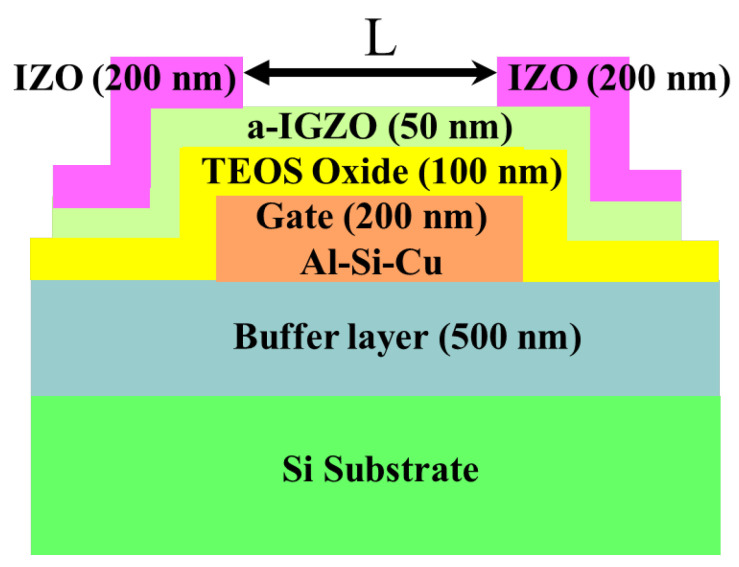
Cross-sectional diagram of the fabricated a-IGZO TFT devices.

**Figure 2 nanomaterials-15-00780-f002:**
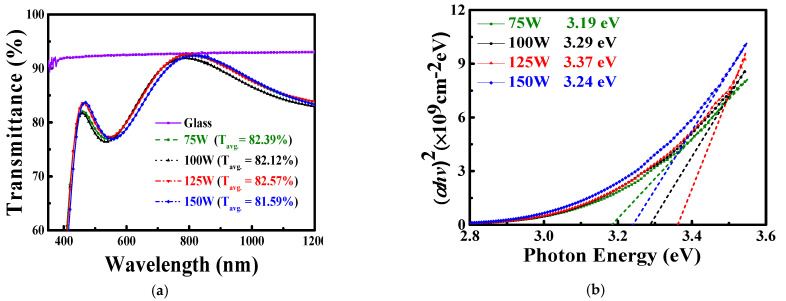
(**a**) Transmittance and (**b**) optical bandgap of IZO films deposited at different sputtering powers after annealing in N_2_ at 300 °C.

**Figure 3 nanomaterials-15-00780-f003:**
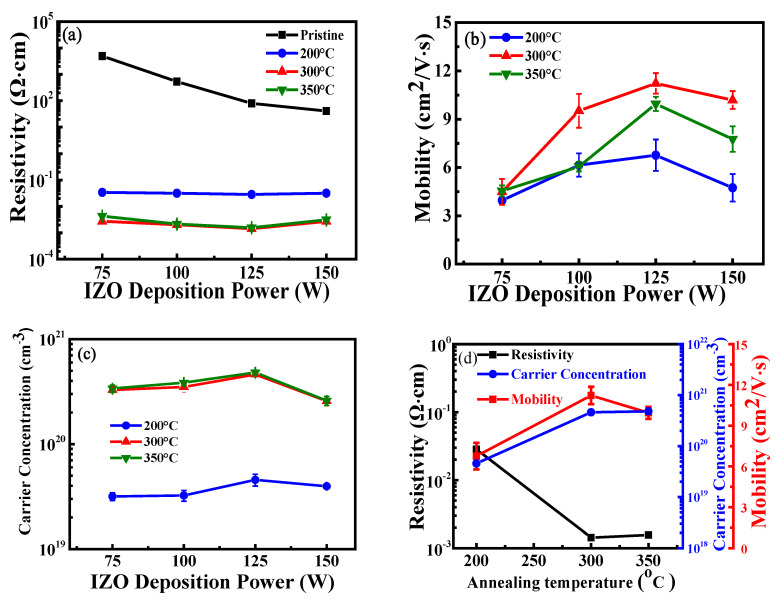
(**a**) Resistivity, (**b**) carrier mobility, and (**c**) carrier concentration of IZO films as a function of sputtering power at different annealing temperatures; (**d**) Hall characteristic of IZO films deposited at 125 W under different annealing temperatures.

**Figure 4 nanomaterials-15-00780-f004:**
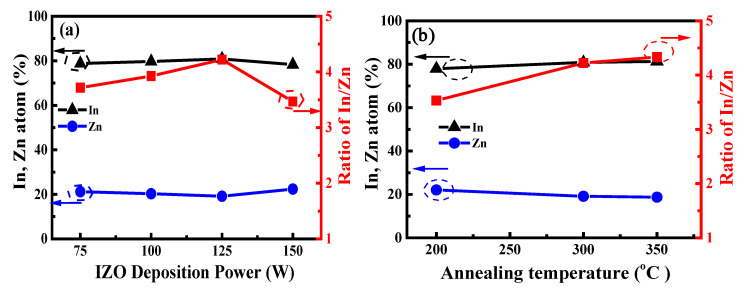
In, Zn, and resulting In/Zn atomic ratios of IZO films (**a**) deposited at various powers and annealed at 300 °C and (**b**) deposited at 125 W as a function of annealing temperature.

**Figure 5 nanomaterials-15-00780-f005:**
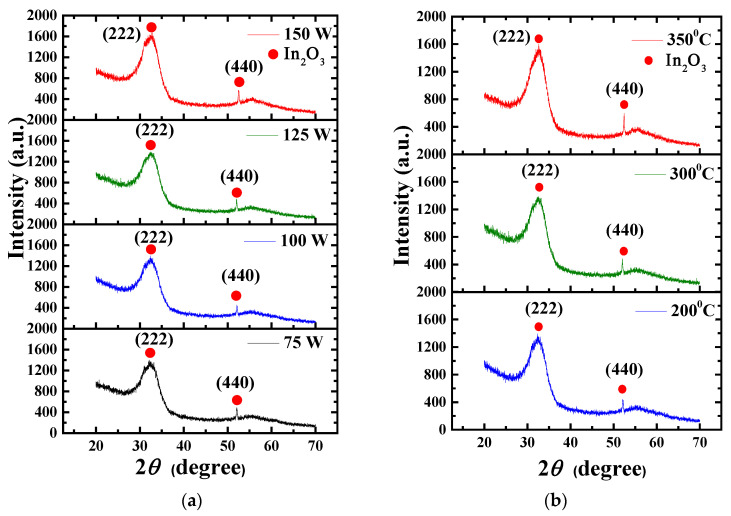
XRD crystalline phase analysis of IZO films: (**a**) at an annealing temperature of 300 °C with different sputtering powers and (**b**) at a sputtering power of 125 W with different annealing temperatures.

**Figure 6 nanomaterials-15-00780-f006:**
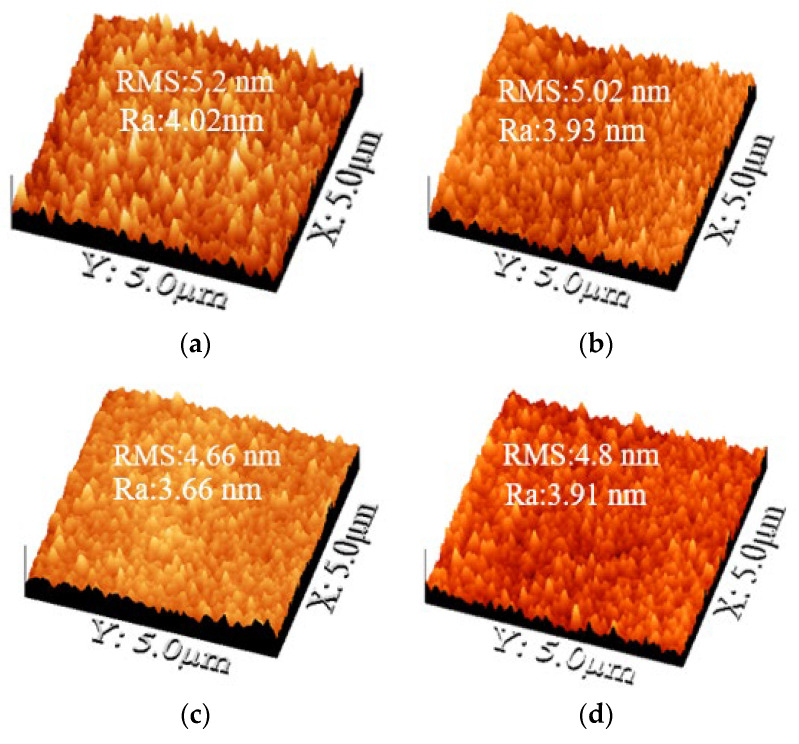
AFM images, RMS roughness, and Ra average roughness of IZO films deposited at (**a**) 75 W, (**b**) 100 W, (**c**) 125 W, and (**d**) 150 W after annealing at 300 °C.

**Figure 7 nanomaterials-15-00780-f007:**
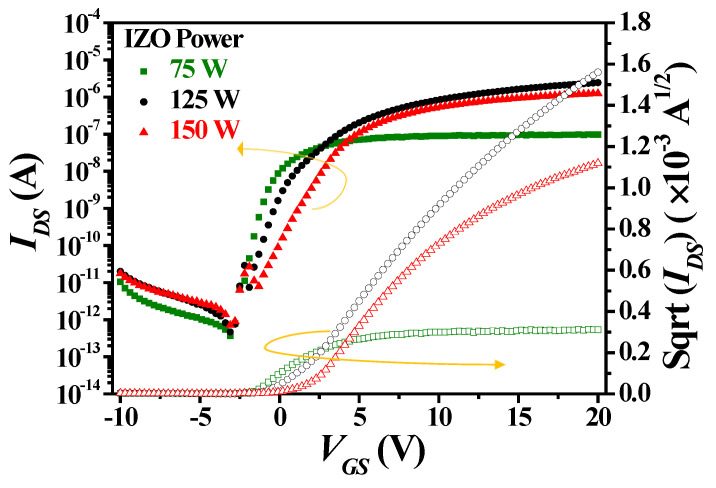
*I_DS_*-*V_GS_* and *I_DS_*^1/2^-*V_GS_* transfer characteristic curves of a-IGZO TFTs at *V_DS_* = 0.1 V and W/L = 400/100 μm with IZO as the S/D electrodes deposited at different powers.

**Figure 8 nanomaterials-15-00780-f008:**
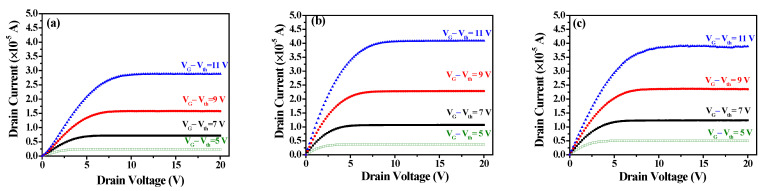
*I_DS_*-*V_DS_* output characteristic curves of a-IGZO TFTs (W/L = 400/100 μm) with IZO as the S/D electrodes deposited at sputtering powers of (**a**) 75 W, (**b**) 125 W, and (**c**) 150 W.

**Figure 9 nanomaterials-15-00780-f009:**
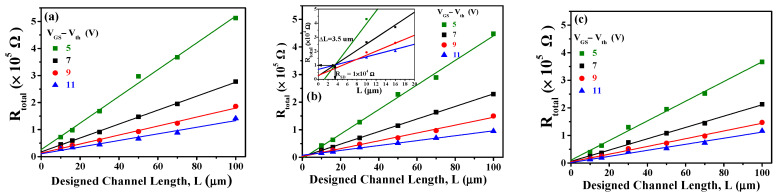
Dependences of *R_total_* on the designed *L* of a-IGZO TFTs with IZO electrodes at different powers: (**a**) 75 W, (**b**) 125 W, and (**c**) 150 W. The inset provides a magnified view of the small L to highlight the point of intersection.

**Figure 10 nanomaterials-15-00780-f010:**
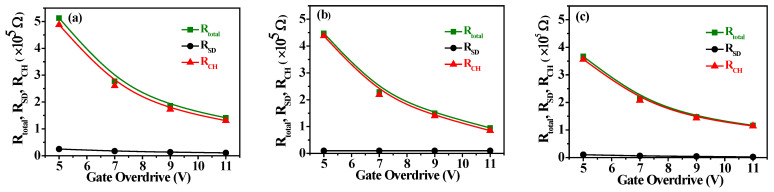
Gate overdrive dependence of *R_total_*, *R_SD_*, and *R_CH_* for a-IGZO TFTs with IZO electrodes annealed at 300 °C under different sputtering powers: (**a**) 75 W, (**b**) 125 W, and (**c**) 150 W.

**Figure 11 nanomaterials-15-00780-f011:**
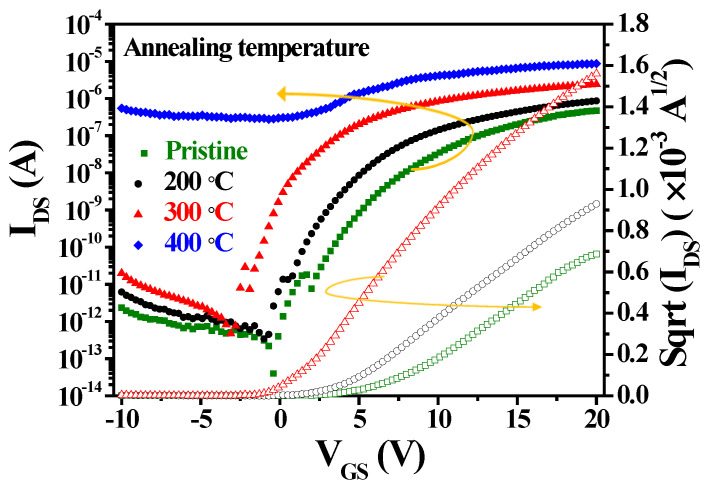
*I_DS_*-*V_GS_* and *I_DS_*^1/2^-*V_GS_* transfer characteristics of a-IGZO TFTs at *V_DS_* = 0.1 V and W/L = 400/100 μm with IZO as the S–D electrodes annealed at different temperatures.

**Figure 12 nanomaterials-15-00780-f012:**
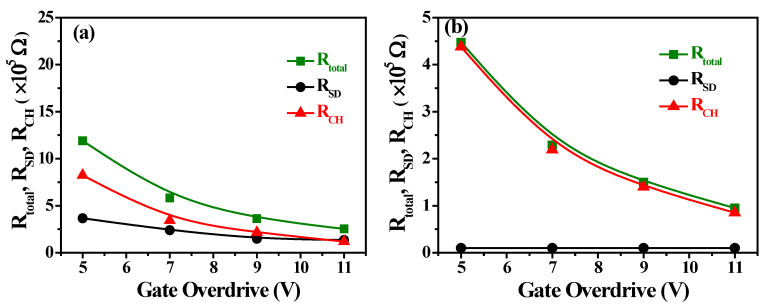
*R_total_*, *R_SD_*, and *R_CH_* plots at different gate overdrive voltages for IGZO TFTs (W/L = 400/100 μm) with IZO as the S–D electrodes at (**a**) 200 °C and (**b**) 300 °C.

**Table 1 nanomaterials-15-00780-t001:** Extracted electrical parameters of a-IGZO TFTs at *V_DS_* = 0.1 V and W/L = 400/100 μm with IZO electrodes at different sputtering powers.

Power (W)	*V_th_* (V)	*S.S.* (V/decade)	*μ_FE_* (cm^2^/V·s)	*I_on_* (A)	*I_off_* (A)
75	−0.24 ± 0.17	0.62 ± 0.11	5.51 ± 0.40	5.92 × 10^−7^	3.7 × 10^−13^
125	0.33 ± 0.04	0.85 ± 0.06	12.31 ± 0.39	2.09 × 10^−6^	7.6 × 10^−13^
150	2.11 ± 0.22	1.22 ± 0.05	9.89 ± 0.98	1.82 × 10^−6^	7.1 × 10^−13^

**Table 2 nanomaterials-15-00780-t002:** Extracted parameters of a-IGZO TFT devices with IZO electrodes annealed at different temperatures.

T_anneal_ (°C)	*V_th_* (V)	*S.S.* (V/decade)	*μ_FE_* (cm^2^/V·s)	*I_on_* (A)	*I_off_* (A)
Pristine	6.6 ± 0.01	1.88 ± 0.16	4.71 ± 0.04	4.69 × 10^−7^	4 × 10^−13^
200	4.08 ± 0.07	1.39 ± 0.03	6.20 ± 0.02	8.76 × 10^−7^	4.5 × 10^−13^
300	0.33 ± 0.04	0.85 ± 0.06	12.31 ± 0.39	2.09 × 10^−6^	7.6 × 10^−13^
400	NA	NA	NA	2.8 × 10^−5^	8.65 × 10^−6^

## Data Availability

The data presented in this study are available on request from the corresponding author.

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
