# Peer review of "Effects of Deposition Power and Annealing Temperature on Indium Zinc Oxide (IZO) Film’s Properties and Their Applications to the Source–Drain Electrodes of Amorphous Indium Gallium Zinc Oxide (a-IGZO) Thin-Film Transistors (TFTs)"

_nanomaterials, 2025, doi:10.3390/nano15110780_

Round 1
Reviewer 1 Report
Comments and Suggestions for Authors
This manuscript investigates the properties of IZO (In-Zn-O) thin films deposited under various sputtering powers and annealing conditions, and their application as source/drain electrodes in a-IGZO TFTs. As a process optimization study based on systematic experiments, the work thoroughly analyzes the correlation between deposition parameters and the resulting film and device performance. While the data acquisition and interpretation are quantitatively solid, the study is somewhat limited in terms of novelty and broader scientific contribution. However, the experimental work is comprehensive, and the conclusions are well-supported by the results. With moderate revisions—particularly in strengthening the discussion and contextualizing the work within existing literature—the manuscript can be significantly improved. Therefore, I recommend a minor revision to enhance its clarity and technical impact.
- The manuscript references “Figure 8,” but no such figure exists. Please correct the figure caption or numbering to ensure consistency throughout the manuscript.
- In the top-right graph of Figure 3, there is an unusual blue "V·s" notation that appears to be a formatting artifact. It is recommended to remove or revise for clarity.
- The overall number of references cited in the manuscript appears to be relatively limited given the scope and context of the study. Considering the substantial body of prior research on IGZO thin films, THz-TDS techniques, and oxide semiconductor characterization, it is recommended that the authors expand the reference list to more comprehensively position their work within the existing literature. Increasing the reference count to around 40 would improve the manuscript’s scholarly depth and help highlight its novelty more effectively. The following papers might be helpful:
Byeon, Gwon, et al. "Recent progress in the development of backplane thin film transistors for information displays." Journal of Information Display 24.3 (2023): 159-168.; Song, Young-Woong, et al. "Doping modulated ion hopping in tantalum oxide based resistive switching memory for linear and stable switching dynamics." Applied Surface Science 631 (2023): 157356.; Lim, Seong-Hwan, Dong-Gyun Mah, and Won-Ju Cho. "Enhancing the Stability and Mobility of TFTs via Indium–Tungsten Oxide and Zinc Oxide Engineered Heterojunction Channels Annealed in Oxygen Ambient." Nanomaterials 14.15 (2024): 1252.; Lee, Seung-Min, et al. "High-Mobility Tellurium Thin-Film Transistor: Oxygen Scavenger Effect Induced by a Metal-Capping Layer." Nanomaterials 15.6 (2025): 418.; Zhao, Mingjie, et al. "The Enhanced Performance of Oxide Thin-Film Transistors Fabricated by a Two-Step Deposition Pressure Process." Nanomaterials 14.8 (2024): 690.; Park, Ji-Min, et al. "Oxide thin-film transistors based on i-line stepper process for high PPI displays." Journal of Information Display 24.2 (2023): 103-108.;
- The manuscript refers to 25 °C as an annealing condition. Since this temperature is essentially equivalent to room temperature, it would be more appropriate to refer to this condition as “pristine” or “as-deposited” rather than “annealed.” Clarifying this terminology would improve technical accuracy.
- While the manuscript provides detailed electrical and structural analyses, it does not address the stability of the TFTs under bias or environmental stress. Including even preliminary stability measurements—such as bias stress tests—would enhance the completeness and practical relevance of the work.
Reviewer 2 Report
Comments and Suggestions for Authors
The experiments seem to be carried out correctly, but there are serious insufficiencies in analysis and argumentation, which is why I recommend a major revision. English is OK, although the style is often lengthy and redundant. In detail:
- Page 4: The optical measurements are probably correct, but I strongly doubt that the optical bandgap can be determined with such a great precision. Especially for 150 W, there are significant deviations between the measurement points and the fits. I assume, that the fitted values are technically correct, but have overlapping error bars.
- Page 4: In a similar way, the average transmittance values are technically correct, but they are so close together that no trend can be predicted. Such small variations can be easily caused by small thickness variations, and I doubt that the layer thickness can be so precisely tailored.
- Page 5: The mobility values in Fig. 3b show error bars in the order of 10% which is reasonable. However, then it does not make sense to give mobility values with a 4-digit-precision like 11.12 cm2/Vs.
- Page 6: This paragraph is a monolithic block over the whole page. Very unpleasant for the reader.
- Page 6: The information about SEM is more or less useless for the reader, and Fig. 4a – d looks very similar. It can be skipped. The authors claim, that grains become “more distinct and denser”, but no numbers are given. And what exactly is a “crystallization phenomenon”?
- Page 6: The authors wrote: “These results indicate that the carrier concentration of the IZO film is proportional to the In/Zn ratio in the composition of the film.”, which is wrong. In fact, both carrier concentration and the In/Zn ratio follow a similar trend, but their relationship is highly non-linear: the carrier concentration changes by orders of magnitude, the In/Zn ratio maybe by a factor of 1.5. In addition, the physical background for this relation is not addressed at all.
- Page 6: The XRD results lacks any serious analysis. The authors only claim that the (440) peak slightly increases with deposition power (which is poorly visible), while an amorphous structure appears at other diffraction angles (which is not very precisely). No peak fit, no information about crystal parameters like grain size or at least the degree of crystallization.
- Page 10: The authors wrote: “…the drain current no longer increases and exhibits current saturation phenomenon.”. The term “current saturation” is used in a wrong context. Current saturation is the progression of I_DS-V_DS curve at higher V_DS values. Correct would be to say that the saturation current no longer increases for a power of 125 W.
- Page 10: The analysis of the relationship between R_total and different channel lengths is highly misleading. At first, you will not be able to determine resistivities with 6-digit-precision as Table 2 suggests. Secondly, R_SD is very difficult to extract, as you subtract two very similar values from each other. Even if possible, R_SD has a very high uncertainty. Because of this, error bars are mandatory.
- Page 10, Fig. 10b: The inset is by far too small.
- Page 11, Table 2: There is probably a major copy error in the column for R_SD for 125 W.
- Page 12: Conclusions are lengthy and redundant. I would focus on the main correlations that really matters (mainly annealing temperature).
Author Response
Please find the enclosed file.
Best regards,

Reviewer 3 Report
Comments and Suggestions for Authors
This manuscript presents the fabrication of an amorphous In-Ga-Zn-O (a-IGZO) thin-film transistor (TFT) device using IZO as source/drain electrodes. The effects of sputtering power and annealing temperature on IZO film properties and their subsequent impact on TFT performance were investigated. While this approach demonstrates potential research value, the provided experimental evidence is insufficient to fully support the claims, and several critical aspects require further clarification and validation. Below are my detailed comments and questions for the authors:
1. The study primarily examines how sputtering power and annealing temperature influence IZO electrode properties and TFT performance. However, the observed changes in TFT performance appear to stem directly from process-induced modifications to the IGZO layer itself (e.g., argon bombardment-induced defects during sputtering power adjustments, direct thermal effects on IGZO during annealing) rather than being explicitly linked to variations in IZO electrode properties. To better align with the stated objective, the manuscript should emphasize how specific property changes in IZO electrodes (e.g., work function, conductivity, interfacial characteristics) modulate IGZO behavior and device performance. The current analysis resembles separate investigations of process parameters on IZO films and TFTs rather than establishing a causal relationship between IZO properties and device characteristics.
2. The number of comparative experimental groups set in the article is insufficient to accurately summarize the patterns of changes in thin-film and device performance caused by variations in process conditions. The article only involves 3 annealing temperatures and 4 deposition powers, which cannot strongly support the summarized patterns. Additionally, according to the test results in the article, the deposition power and annealing temperature have limited impact on the performance of IZO electrodes.
3. Formatting Issues:
Figure 4: Subfigure label "(b)" is missing in the caption.
Figure 6: Extraneous line should be removed.
Figure Duplication: Two distinct figures are erroneously labeled as "Figure 7."
Author Response

(The authors gave the same response as above.)

Round 2
Reviewer 1 Report
Comments and Suggestions for Authors..............
Author Response
Response to Reviewer 1's Comments:
Comments and Suggestions for Authors
- The English could be improved to more clearly express the research.
Reply: Response to Reviewer:
We thank the reviewer for pointing out the need to improve the clarity of the English in our manuscript. In response, we have thoroughly revised the entire manuscript to enhance the readability and ensure that the research objectives, methodology, results, and conclusions are expressed more clearly and precisely. The revisions in the document have been highlighted in yellow.
Reviewer 2 Report
Comments and Suggestions for Authors
The authors made great effort to improve the manuscript, and it can be now published in its present form. The only remaining issue is that the inset of Fig. 9b is still too small. As the inset is just a magnification of fig. 9b, I suggest to delete it and to increase the visibility of the cross marking the intersection. Now it’s a pale thin line that is difficult to see.
Reviewer 3 Report
Comments and Suggestions for Authors
The authors have thoughtfully addressed the initial concerns raised in the review, demonstrating a clear commitment to strengthening the manuscript. Their revisions effectively establish a causal relationship between IZO electrode properties and TFT performance, enhance the statistical rigor of experimental results, and resolve formatting issues. The response is thorough, with specific references to revised figures and analyses that directly address each comment.
